# In Situ SEM Study of the Micro-Mechanical Behaviour of 3D-Printed Aluminium Alloy

**Eugene S. Statnik** [1] , **Kirill V. Nyaza** [2], **Alexey I. Salimon** [1,*], **Dmitry Ryabov** [2] and **Alexander M. Korsunsky** [3,1]

1   HSM Lab, Center for Energy Science and Technology, Skoltech, Moscow 121205, Russia; eugene.statnik@skoltech.ru (E.S.S.); a.korsunsky@skoltech.ru (A.M.K.)
2   Light Materials and Technologies Institute, UC RUSAL, Moscow 121096, Russia; mobiad@yandex.ru (K.V.N.); dmitriy.ryabov2@rusal.com (D.R.)
3   MBLEM, Department of Engineering Science, University of Oxford, Oxford OX1 3PJ, UK
*   Correspondence: a.salimon@skoltech.ru

**Abstract:** Currently, 3D-printed aluminium alloy fabrications made by selective laser melting (SLM) offer a promising route for the production of small series of custom-designed support brackets and heat exchangers with complex geometry and shape and miniature size. Alloy composition and printing parameters need to be optimised to mitigate fabrication defects (pores and microcracks) and enhance the parts' performance. The deformation response needs to be studied with adequate characterisation techniques at relevant dimensional scale, capturing the peculiarities of micro-mechanical behaviour relevant to the particular article and specimen dimensions. Purposefully designed Al-Si-Mg 3D-printable RS-333 alloy was investigated with a number of microscopy techniques, including in situ mechanical testing with a Deben Microtest 1-kN stage integrated and synchronised with Tescan Vega3 SEM to acquire high-resolution image datasets for digital image correlation (DIC) analysis. Dog bone specimens were 3D-printed in different orientations of gauge zone cross-section with respect to the fast laser beam scanning and growth directions. This corresponded to the varying local conditions of metal solidification and cooling. Specimens showed variation in mechanical properties, namely Young's modulus (65–78 GPa), yield stress (80–150 MPa), ultimate tensile strength (115–225 MPa) and elongation at break (0.75–1.4%). Furthermore, the failure localisation and character were altered with the change in gauge cross-section orientation. DIC analysis allowed correct strain evaluation that overcame the load frame compliance effect and helped to identify the unevenness of deformation distribution (plasticity waves), which ultimately resulted in exceptionally high strain localisation near the ultimate failure crack position.

**Keywords:** RS-333 alloy; SLM 3DP; in situ SEM tensile testing; DIC analysis; *Ncorr*

## 1. Introduction

Following a period of rapid development since the early 2000s, the additive CAD/CAM technology of selective laser melting (SLM) 3D printing of metal alloys was expanded to include the use of aluminium alloys in the 2010s [1–3]. Important process aspects such as the alloy composition, laser scanning rate and post-processing parameters were systematically investigated to achieve desirable mechanical performance [4–6]. Developed on the basis of the widely used near-eutectic casting Al-Si alloys, the newly formulated 3D-printable Al-Si-Mg alloys facilitate the fabrication of parts with low porosity through ensuring extended solidification times and good flowability, leading to superior casting and printability behaviour and low susceptibility to hot cracking. The improvement in the liquid-phase viscosity by means of doping Al-Zn-Mg-Cu high-strength alloys with secondary alloying elements (Zr, Mn, Fe, Co and others) reduces the propensity of undesired defect assemblies (pores and coarse dendrite grains) down to acceptable levels [4]. Moreover, transition elements such as Zr, Ti or Mn have strong grain-refining effects, leading to significant improvements in the resistance to hot cracking due to the decreased size of dendrites,

which allows better liquid metal support within solid–liquid regions during solidification. The introduction of such micro-alloying elements even in the form of nano-sized particles is considered to be an efficient mean of improving printability [4]. Nevertheless, other phenomena must be taken into consideration that occur at hierarchically different dimensional structural levels. These include metallurgical aspects (formation of oxide inclusions, liquation and inherited component inhomogeneity) and thermo-mechanical processes (cracking and residual stresses caused by strong thermal gradients and solidification shrinkage). Control over these phenomena is required in order to optimise the mechanical and functional performance of 3D-printable aluminium alloys and components to attain high geometric fidelity of 3D-printed parts [7].

Technologically, the more complex printable aluminium alloys present a number of advantages with respect to their traditional counterparts for niche applications where longer production time per article is tolerable. Prototypes, single units or small batches of miniature parts having complex geometry for use in fine mechanics applications, computer components and gadget hardware and robotics currently represent a clear scope for SLM technology evolution [8].

Additive manufacturing technologies can find application when specific combinations of properties are sought, such as for miniature heat exchangers of least mass or volume when both high strength and thermal conductivity are required. The performance indices to be minimised, according to Ashby [9], are $\frac{\rho}{\sigma_y^2 \cdot \lambda}$ and $\frac{1}{\sigma_y^2 \cdot \lambda}$, respectively, where $\rho$ is density, $\sigma_y$ is yield strength and $\lambda$ is the thermal conductivity.

Traditionally, components such as heat exchangers are fabricated of 1XXX, 3XXX (Al-Mn) or 6XXX (Al-Mg-Si) series alloys. On the other hand, new aluminium alloys purposefully optimised for 3D printing possess both higher strength (compared to traditional AlSi$_{10}$Mg alloy used in 3D printing) and improved thermal properties. Though aluminium-based materials are generally expected to have high thermal conductivity [10], favourable to produce various parts for heat exchangers, 3D-printable AlSi$_{10}$Mg alloy manifests moderate heat conductivity. Moreover, 6XXX alloys possess rather high strength and acceptable thermal conductivity, as shown in Figure 1, but alloys such as 6061 are prone to cracking during the printing process [11], prompting researchers to seek new alloy formulations to adopt for AM processes. The composition of the 3D-printable RS-333 alloy from the Al-Mg-Si family considered in this paper was tuned by design to improve castability and thereby to provide a substitute for 6061 aluminium alloy in heat exchange applications.

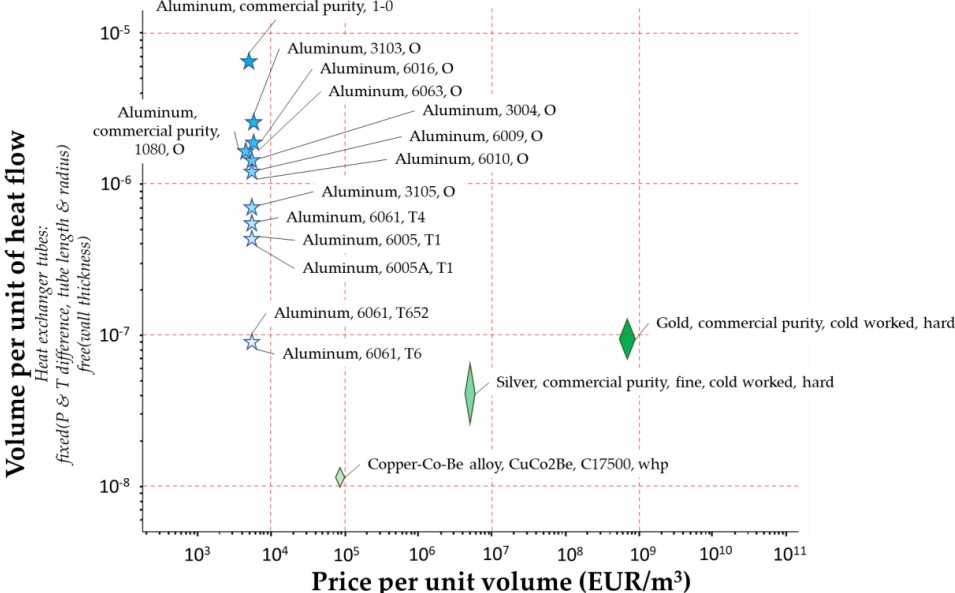

**Figure 1.** Comparison of some materials used in heat exchanging applications. Chart and data from CES EduPack 2019, Granta Design Limited, Cambridge, UK, 2019 [12].

Mechanical microscopy methods that couple optical and/or electron microscopy with mechanical testing and advanced digital image analysis have been rapidly developing in recent decades [13–17]. These methods are particularly suitable for in situ mechanical testing of near-net-shape miniature and thin parts inside an SEM chamber, since the deformation and fracture behaviour can be visualised and studied at high resolution, readily applicable to digital image correlation (DIC) analysis to reveal the peculiarities of deformation at the micrometre scale. Moreover, SLM 3D-printing of miniature fine mechanics parts having slim cross-sections occurs under the cooling and solidification conditions that differ from those for conventional engineering parts.

We report our findings in a systematic study of the micro-mechanical behaviour of tensile samples made from 3D-printable alloy RS-333. The acquisition of high-resolution SEM images was synchronised with in situ tensile testing of dog bone samples 3D-printed in different relative orientations of the laser scanning and growth directions with respect to the sample shape. DIC analysis was used to map the strain distribution and thus to trace the localisation of strain in the vicinity of the major crack. The interpretation of the experimental findings suggests a correlation between the duration of the effective cooling period for elementary added material volumes, on the one hand, and the mechanical performance on the other.

## 2. Materials and Methods

The powder of RS-333 (Al-3Si-0.5Mg) alloy was supplied by "Valcom-PM" Ltd. (Volgograd, Russia). The powder was produced by a nitrogen atomisation method. Particle size varied in the range of 20–63 μm with D50 = 42 μm according to the data from laser particle size analysis performed with ANALYSETTE 22 Nano Tec (Fritsch, Idar-Oberstein, Germany). The SEM appearance of the RS-333 powder is presented in Figure 2.

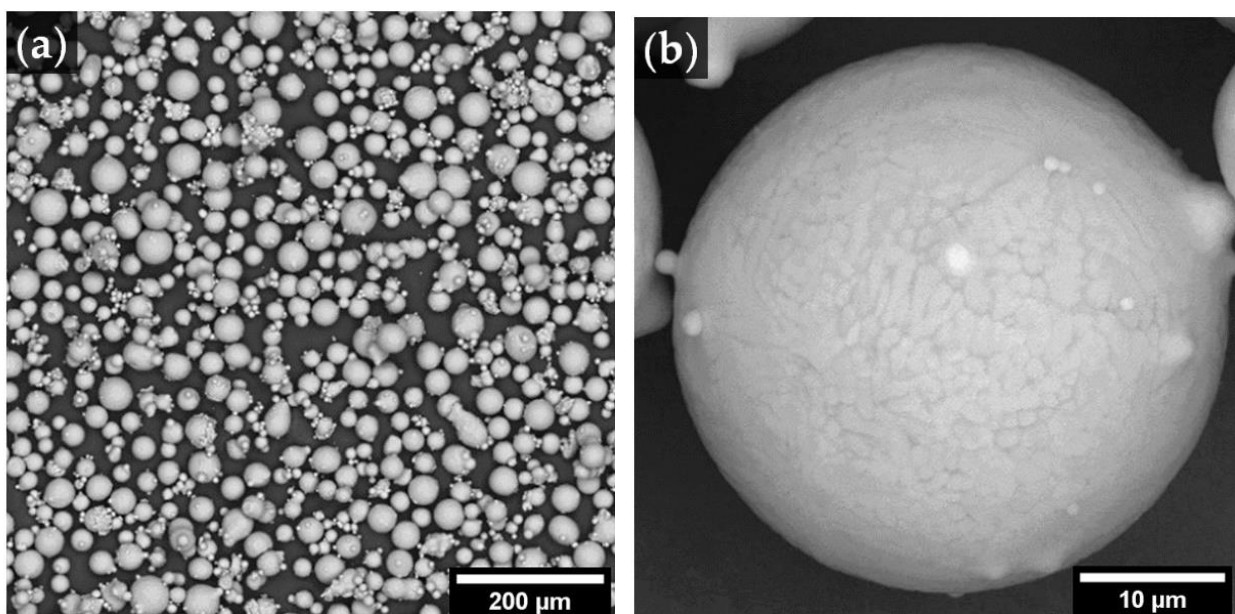

**Figure 2.** SEM images of RS-333 powders: (**a**) general view and (**b**) single particle microstructure.

A well-known challenge in SLM concerns 3D-printing of highly reflective aluminium alloy powders that possess inherent reflectivity of up to 90% in the IR range at ~1-μm wavelength. This was successfully overcome by the feedstock supplier using proprietary methods of powder surface modification (roughening) to allow this additive manufacturing technology to be applied at industrial scale.

The 3D-printing SLM process (powder bed fusion) was carried out using an EOS M290 SLM printer (Germany) equipped with a 400-W Yb fibre laser unit with a wavelength of 1075 nm. Argon of high purity was used during printing to avoid oxidation of the

powder and the melt. Scanning speed used was 800 mm/s at the laser power of 370 W. Layer thickness was set to 30 µm. A post-printing heat treatment (aging) at 160 °C for 12 h to release residual stress and to promote precipitation hardening was applied to all printed samples. All samples were printed using "core" parameters without special skin exposure for further surface treatment.

In this research, ASTM E8 standard was used for sample geometry. Flat dog bone specimens having thickness of ca. 1 mm and gauge length of 10 mm were printed in a layer-by-layer process. Sets of at least 3 samples of each different orientation of main gauge axis with respect to the fast (X) and slow (Y) laser scanning directions and the growth axis (Z), as shown in Figure 3, were fabricated to study the influence of printing orientation on the mechanical response under tension. In our notation, the first character corresponds to the axis aligned with gauge length, the second to the axis aligned with gauge width. Supports used during printing were mechanically machined off to obtain samples of nominal dimensions.

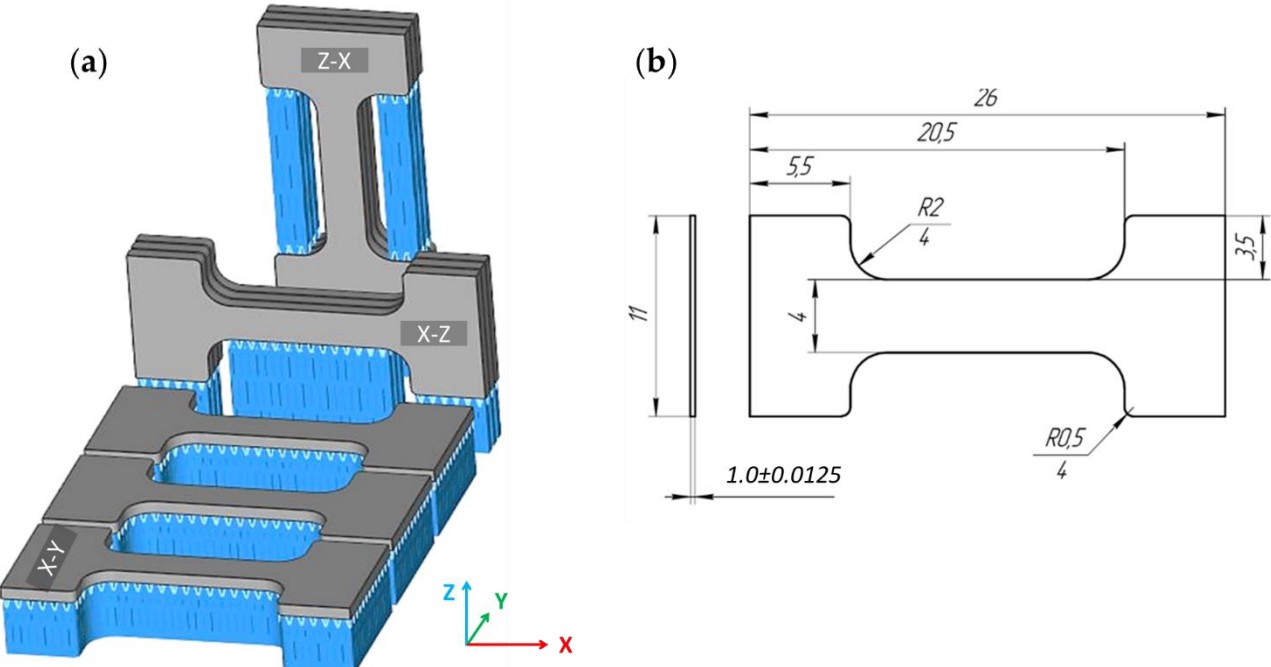

**Figure 3.** Test samples (**a**) of varying orientation during printing process, X-Y, X-Z and Z-X, shown from front to back, and (**b**) the nominal sample shape and dimensions.

Microstructure studies were performed after grinding and polishing using Struers laboratory equipment of as-printed sample cross-sections with no additional chemical etching. Optical microscope Zeiss Axio Observer 7 and scanning electron microscope Tescan MIRA 3 LMH (Tescan Company, Brno, Czech Republic) were applied to quantify residual pores and visualise internal fine structure, respectively. As a rule, the analysis of 10 fields of cross-sections in Z direction was carried out for porosity assessment.

In situ mechanical testing was facilitated through the use of a Deben Microtest 1-kN testing stage placed in the chamber of a Tescan Vega 3 SEM (Tescan Company, Brno, Czech Republic). The testing stage was operated under control of a Python code [18] to synchronise the mechanical loading (conducted at a permanent crosshead speed of 0.2 mm/min) with the acquisition of SEM images, as illustrated in Figure 4.

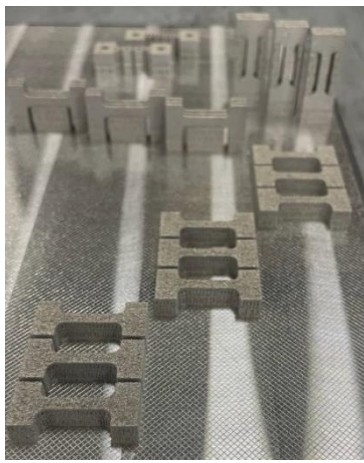 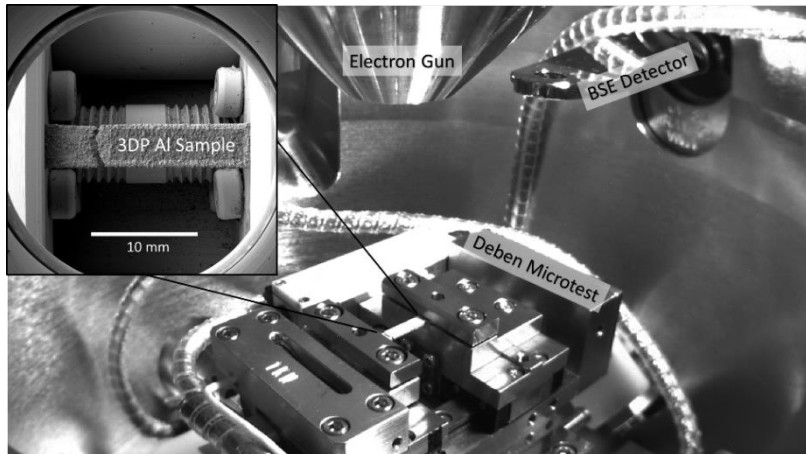

**Figure 4.** Appearance of 3D-printed specimens (**left**) and principal layout of experimental setup for in situ SEM study of mechanical response (**right**).

SEM images were acquired at the rate of 22 s per image in the secondary electron regime using 30 kV voltage and beam spot size of 400 nm. Videos S1–S6 of the specimen deformation and fracture process can be found in the Supplementary Materials.

Digital image correlation analysis with the use of open-source Matlab-based software *Ncorr* [19] was applied to the series of SEM images (up to ~40–50 images depending on the dataset acquired till sample break) to map the distribution of displacement and strain with subpixel resolution. The DIC algorithm compares two digital images (arrays containing digitalised intensity values) aiming to find the best match between pixel subsets. After the determination of the centre positions of corresponding pixel subsets, the displacement and eventually strain fields are calculated.

The definition of regions of interest and pattern quality (density of distinguishable surface features) affects both robustness and computing time in DIC analysis being performed with help of *Ncorr*. As seen in Figure 5, the pattern quality was satisfactory in terms of contrast and surface feature density. In this paper, 80% of the gauge area was used for DIC analysis, as illustrated in Figure 6.

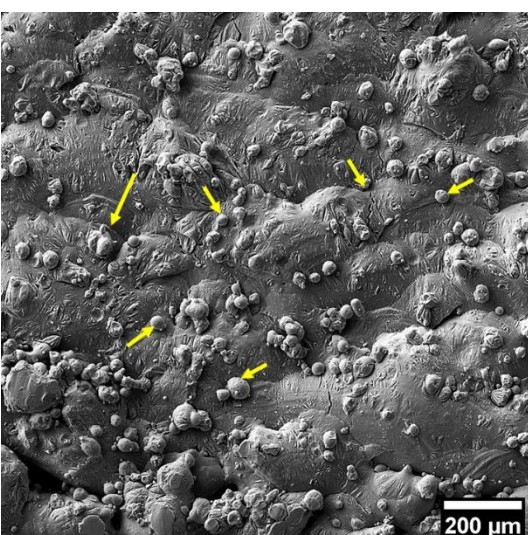

**Figure 5.** Appearance of surface of 3D-printed RS-333 alloy with the indicated (yellow arrows) surface features.

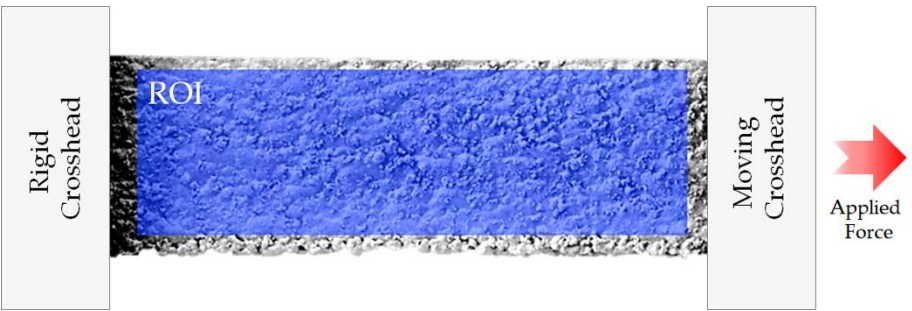

**Figure 6.** Illustration of typical ROI (Region Of Interest) selection for captured SEM image.

An important issue of elongation calibration at loading curves needs to be addressed when elastic moduli and true strains are sought to be derived from raw data acquired from a testing stage such as the Deben Microtest 1 kN. Crosshead displacement data are returned from a Linear Variable Differential Transformer (LVDT) sensor. Despite the high accuracy of the LVDT sensor, several factors must be taken into account when loading curves are analysed, namely (1) the effect of machine load frame compliance on the apparent deformation, and (2) the non-uniformity of the plastic and total strain (rate) during tensile deformation [20]. In the absence of suitable correction, the apparent deformation response of each sample depends on the stiffness of the test machine used. There are several methods of so-called specimen–machine coupling effect determination that allow the extraction of true specimen deformation data from the test results. The most straightforward method is the total deformation analysis method, in which the crosshead compliance correction for the displacement values is performed using the following formulae:

$$u_{total} = u_{sample} + u_{machine};$$
(1)

$$u_{sample} = u_{total} - u_{machine} = u_{total} - \frac{F}{k_{machine}},$$
(2)

where

$u_{sample}$ is true sample displacement (mm);
$u_{total}$ is the displacement measured during the test (mm);
$F$ is the force measured during the test (N);
$k_{machine}$ is load frame stiffness (N/mm);
$1/k_{machine}$ is load frame compliance (mm/N).

In the present study, a special "stiff" Deben calibration specimen of SS316 steel with cross-section of 300 mm$^2$ and hence negligible deformation under load was used to obtain the calibration loading curve and derive $k_{machine}$ for the particular Deben Microtest 1-kN device.

On the other hand, DIC analysis is able to return strain maps in the region of interest (ROI) as well as the average strain along the main axis at the gauge length. The example of the selected ROI for each sample was similar in accordance with Figure 6. The longitudinal strain averaged over the ROI was used to plot the corresponding loading curve depicted in Figure 7.

In Figure 7, the data corrected for machine compliance in comparison with DIC analysis results are illustrated for a typical sample. It is apparent that DIC analysis yields the most reliable values of Young's modulus, so that, hereinafter, true strains determined by means of DIC analysis are used in the analysis.

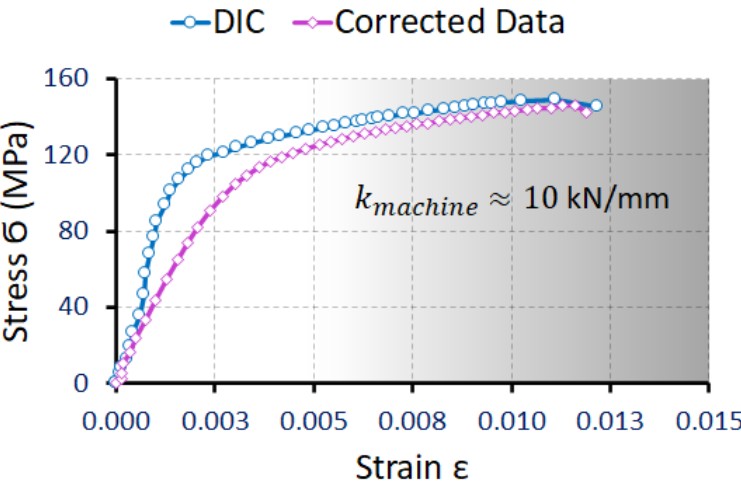

**Figure 7.** Illustration of sample stress–strain curves obtained from DIC analysis with the testing stage data corrected to account for the testing rig frame compliance.

## 3. Results and Discussion

### 3.1. Microstructure Studies

Selected parameters of the powder bed fusion process applied for the designed RS-333 powder result in the formation of a material structure with rather low porosity and having no detectable internal hot cracks, which indicates good service characteristics. Porosity calculated from the analysis of optical microscopy images as shown in Figure 8 is as low as 0.3%, which is typical for 3D-printed aluminium alloys [21].

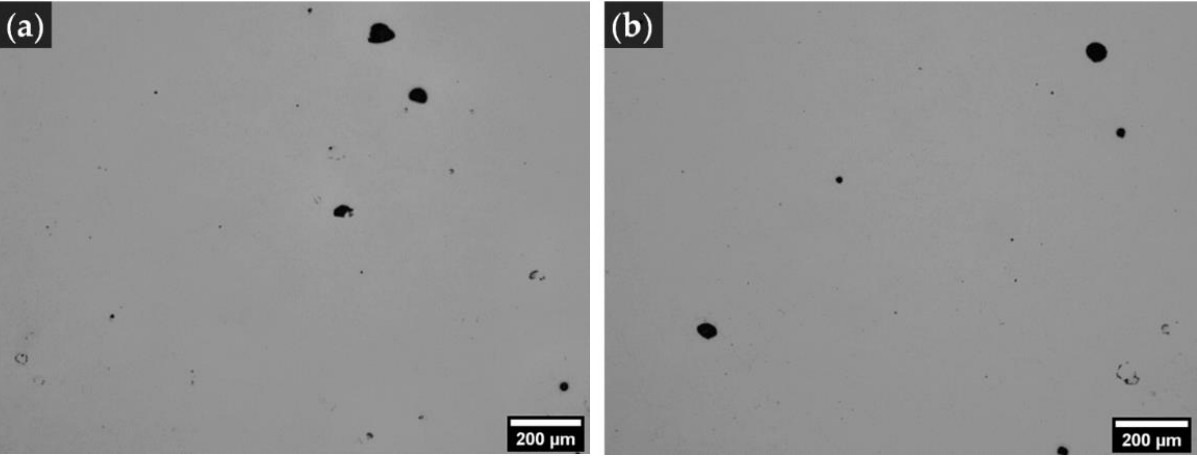

**Figure 8.** Representative images (**a**,**b**) of porosity within SLM RS-333 aluminium alloy parts taken in the X-Z sectional plane orientation.

SEM images of the structure formed in RS-333 alloy samples as a result of SLM 3D-printing and aging are represented in Figure 9. Si phase precipitates in the aluminium matrix are mainly represented by the curved and dashed chains, and some smaller equiaxial particles are also noticeable, suggesting that Si in RS-333 alloy is prone to some spheroidisation at the aging temperature applied. The improvement in heat conductivity takes place due to the decomposition of oversaturated solid solution. Aging temperature of 160 °C is, however, insufficient to complete the spheroidisation, eventually giving an optimal combination of high heat conductivity and mechanical performance.

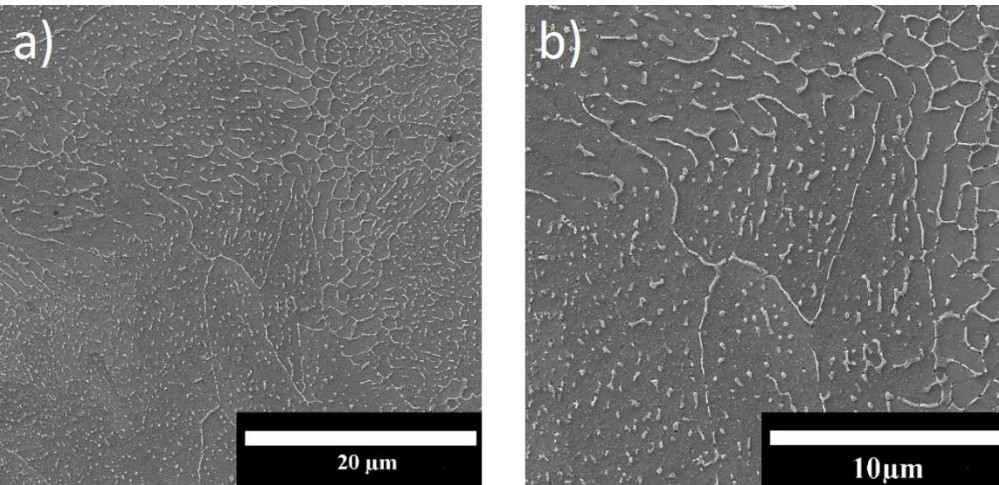

**Figure 9.** SEM images of RS-333 alloy microstructure following SLM printing and aging: (**a**) low and (**b**) high magnifications.

Surface appearance of an X-Z sample is represented in Figure 10, revealing relatively large (200–400 µm) and smooth clusters of molten material decorated with wrinkles and rare cracks, as well as the inclusion of non-molten powder particles. The rest of the supports are visible at the bottom edge of the gauge, where the clusters are apparently coarser and appear to be more separated with voids than in the middle zone of the gauge. These locations and support-free surfaces did not undergo stable continuous laser scanning and powder fusion. In contrast, the metallographic examination of the specimens' core (stable laser scanning and powder fusion) revealed residual porosity only (Figure 8) and no traces of unfused powder.

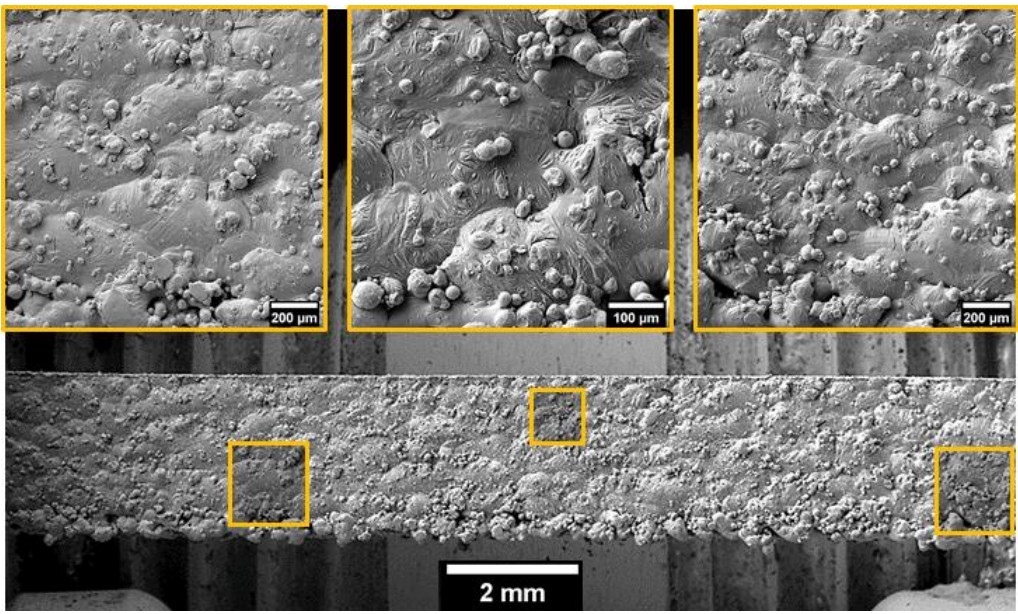

**Figure 10.** Surface appearance of X-Z sample with structure specification.

### 3.2. Mechanical Performance versus Sample Orientation

Mechanical performance is significantly affected by the growth orientation during SLM 3D-printing, as illustrated by the data represented in Figure 11 and Table 1. This technology is intrinsically complex and affects the structure (and mechanical performance) at hierarchically scaling—from sub-micrometre up to millimetre—dimensional levels, making the interpretation and understanding especially challenging in terms of intensive structure characterisation.

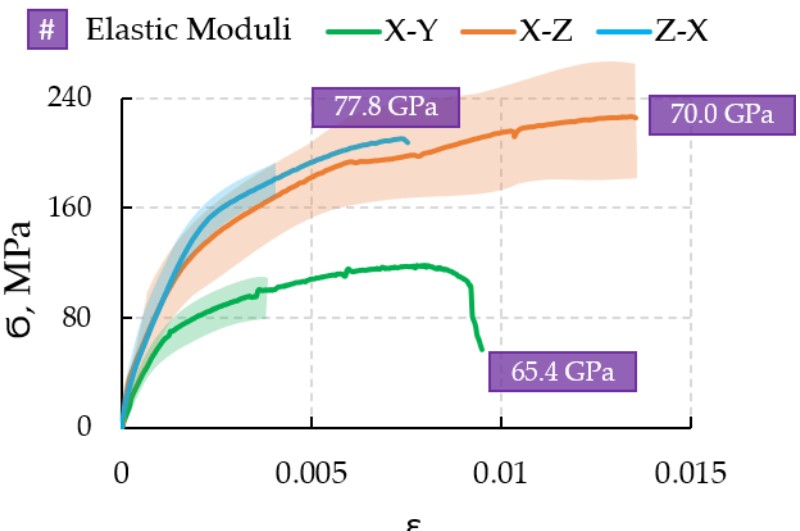

**Figure 11.** The typical stress–strain curves obtained for different sample printing orientations with the Young's modulus also indicated.

**Table 1.** Mechanical performance of RS-333 alloy samples with different sample orientations.

| Sample Orientation | Young's Modulus, GPa | Yield Strength at 0.2% Strain, MPa | Ultimate Tensile Strength, MPa | Elongation at Rupture, % | Path Time, s | Layer Time, s |
|---|---|---|---|---|---|---|
| X-Y | 65.4 | 81 | 115 | 0.95 | 0.03 | 0.30 |
| X-Z [1] | 70.0 | 132 | 227 | 1.36 | 0.03 | 0.10 |
| Z-X | 77.8 | 150 | 210 | 0.75 | 0.005 | 0.015 |

[1]—sample break takes place outside of gauge zone, i.e., in the rounded zone between gauge and clamp.

A conceptual model can be put forward to highlight the aspect of thermal history. Cooling rate from melting temperature and peak temperatures at reheating from subsequent paths and layers in a certain material micro-volume play an important role in the structure formation, including at least the following: (a) solid solution composition (and Si oversaturation) after solidification; (b) ultimate grain size; (c) development of aging processes—volume fraction and morphology of Si precipitates.

Taking into account that the area of the gauge cross-section directly depends on the sample orientation, one can easily estimate the characteristic times needed to print a path in plane (path time) and a layer (layer time) using the following formulae:

$$\tau_{path} = \frac{l_{path}}{v_{scan}}, \tag{3}$$

$$\tau_{lay} = \frac{S_{cs}}{d_{spot} \cdot v_{scan}}, \tag{4}$$

where $\tau_{path}$—path time (s)
$\tau_{lay}$—layer time (s);
$l_{path}$—path length (along fast scanning direction) in gauge zone (mm);
$S_{sc}$—area of cross-section in gauge zone (mm$^2$);
$d_{spot}$—diameter of fusion zone (scaled with laser power) (mm);
$v_{scan}$—scanning speed (mm/s).

The estimations listed in Table 1 were calculated for $d_{spot} = 300$ μm and $v_{scan} = 800$ mm/s in this research. The cooling rate for a particular micro-volume correlates with its particular position inside the cross-section as well as the geometry and area of the solidified cross-section, since both temperature gradient and the area of surrounding heat sinking zone depend on the latter characteristics. More accurate calculations of cooling rates can

be modelled with a multi-physics FE simulation [22], but for qualitative considerations, cooling rate can be taken as a reverse function of path and layer time. Thus, the X-Y orientation with the highest cooling rate and less frequent reheating events ultimately brings the studied alloy to the structure state corresponding to solid solution heat treatment followed by natural aging. The smallest grain size is also expected for this orientation.

In contrast, the Z-X sample orientation with the lowest cooling rate and frequent reheating events tends to form the structure of an annealed (with obvious recrystallisation) or overaged solid solution with coarse grains. The X-Z orientation was found to be intermediate (and, perhaps, optimal) in terms of structure (solid solution heat treatment and artificial aging) and mechanical performance.

On the other hand, phenomena of another nature, such as the generation of residual stresses, chemical inhomogeneity and dimensional unevenness, are to be taken into account as factors which may predominate over microstructural effects.

This conceptual modelling may satisfactorily explain the correlation between SLM 3D-printing build orientation and mechanical performance attributes such as hardness, yield and ultimate tensile strength and elongation at rupture. These properties are mainly dominated by phase composition and structure morphology—grain size and orientation (texture). Few recent studies have been focused on the systematic analysis of aluminium [21], nickel [23], titanium alloys [24] and stainless steel [25].

Variations in mechanical properties against build-printing orientations were observed in many articles for different aluminium-based alloys prepared by the SLM technique [26]. For instance, Li X. [27], Ch S.R. [28] and Tang M. [29] investigated an $AlSi_{10}Mg$ alloy and achieved the same results as in our study; namely, samples manufactured in the vertical direction (Z-X) showed higher values of ultimate tensile strength than specimens built in the horizontal direction (X-Y). However, there are other studies where samples printed with equal conditions and composition demonstrated the inverse relation [30,31]. Moreover, this stochastic behaviour of mechanical characteristics was found during tensile tests of samples with other alloy compositions such as $AlSi_{10}$, A356 ($AlSi_7Mg_{0.3}$) and A357 ($AlSi_7Mg_{0.7}$) [27].

The issue of Young's modulus is more challenging to understand for aluminium alloys since these materials are almost elastically isotropic, so that the directional dependence of stiffness cannot be explained by texture variation. However, it should be noted that, similarly to our results, it has been shown that Young's modulus varies significantly within the range of 63–72 GPa for SLM 3D-printed $AlSi_{10}Mg$ alloy [31].

We believe that a strong variation in the elastic modulus for Al-Si-Mg is likely to be related to the peculiarities of the spatial arrangement of precipitates and nanometre-sized pores. However, detailed elucidation of these intricate relationships requires systematic studies by means of high-resolution techniques such as EBSD and FIB tomography, which are currently being performed by different research groups, including the present authors, in order to elucidate the contribution of grain orientation and porosity to the elastic behaviour of RS-333.

### 3.3. DIC Insight into Strain Distribution

It is demonstrated in Figures 12 and 13 and Figure A1 that DIC analysis mapping reveals the inhomogeneous distribution of strains even at the earliest stage of tension when a sample is macroscopically elastic. Strain distribution can be described as a wave with a wavelength along longitudinal axes of approximately 2 mm, which may suggest the influence of technological factors to be optimised for more strict quality control.

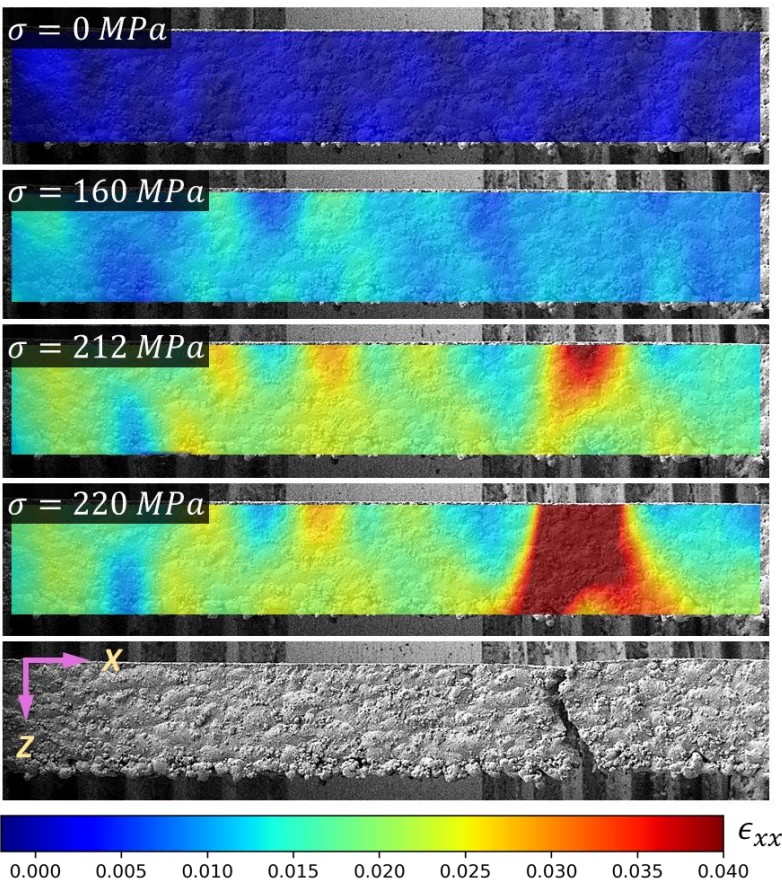

**Figure 12.** Demonstration of 2D DIC processing technique, namely continuous strain localisation tracking.

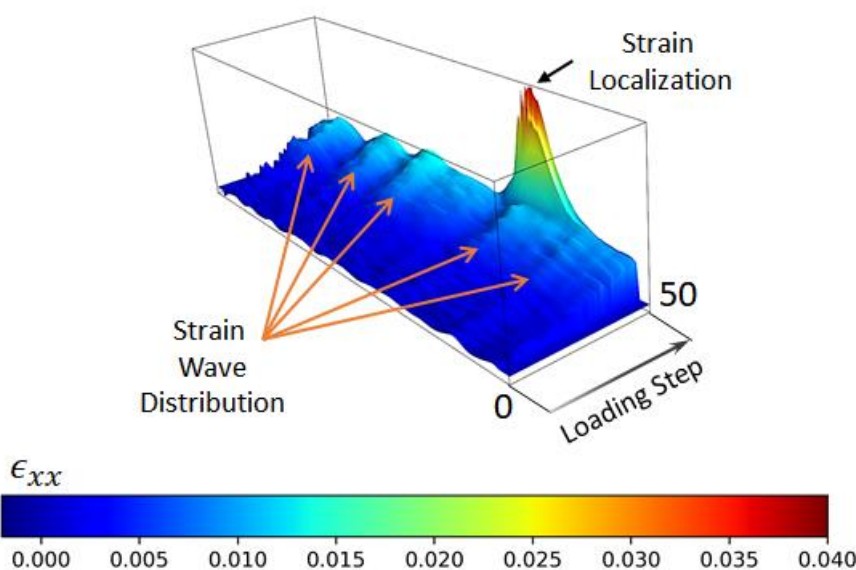

**Figure 13.** 3D DIC result representation with indicated strain wave-like distribution behaviour.

Localisation of strain in the vicinity of future cracks happens drastically and immediately before the break, which may be explained in stochastic terms, such as an inherited

crack, a pore or overgrown grain or a non-metallic inclusion. In any case, almost no necking was noticed, leaving the issue of survivability unresolved.

It is worth noting that some samples having X-Z built orientation showed that the localisation of strains and further rupture occur in the rounded zone between the gauge and clamp. In this zone, the transition from long gauge zone to short clamp zone takes place at longitudinal laser scanning. There is a narrow zone where abrupt changes in path length, layer time and cooling rate occur, causing the unevenness of the temperature field and residual stresses at further overall cooling (temperature conditions are to be simulated with computer modelling to quantify this effect). We suggest that the temperature and time of post-processing annealing were not optimised for the release of these stresses. This gives very useful guidance for good practice in SLM 3D-printing.

### 3.4. Fracture Surface Appearance

The appearance of the fracture surface in Figure 14 suggests that the RS-333 alloy manifests some ductility during the initiation and growth of a crack. The crack line occurs at an angle around 60 degrees to the longitudinal axis; however, smaller and larger angles were also detected.

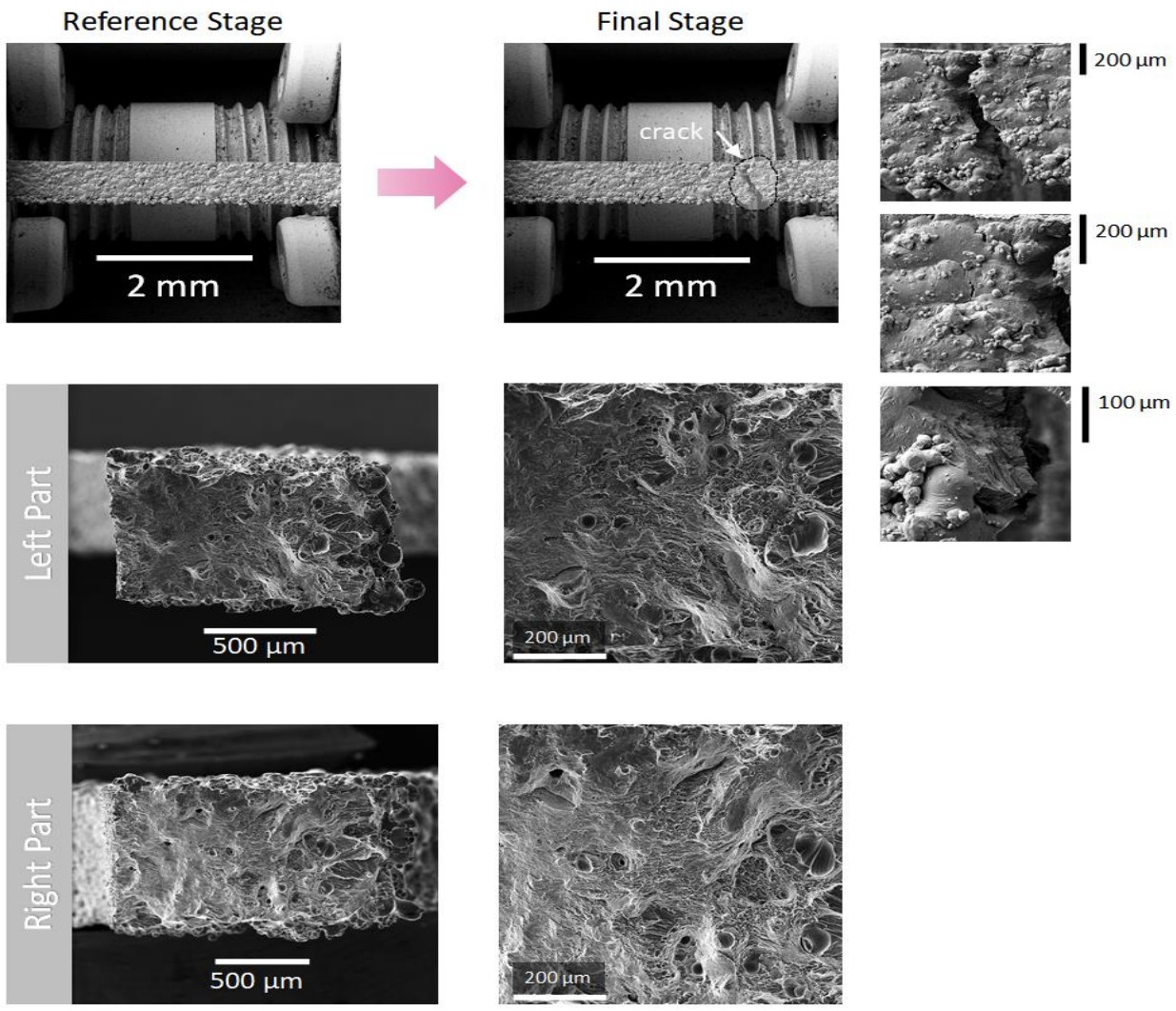

**Figure 14.** Appearance of fracture surface of an X-Z specimen.

The fracture surface (Figure 14, from left to right) has relatively smooth zone with few round pits, followed by a zone with elongated fringe features and finally a zone with coarse pit features at the edge, where the rest of the supports were noticed (Figure 10). It can be seen in Figure 12 that the crack is initiated from the flat edge zone opposite to the edge with coarse clusters and the rest of the supports. It seems that the crack starts to grow in accordance with ductile mechanisms (smooth zone) and terminates as a brittle crack (rough zone with elongated fringe features) when substantial stress concentration is reached. The initiation of a crack happens in the zone where sample growth ends and material consolidation evolves, with no remelting and reheating. On the other hand, this zone has a much firmer substrate (specimen body) than at the opposite stage, where only supports are present. Deeper structural investigations are required to reliably identify the main physical phenomena governing the fracture behaviour.

## 4. Conclusions

For dog bone specimens, SLM 3D-printed Al-Si-Mg alloy RS-333 reveals a significant interrelation between the specimen orientation and mechanical tensile performance, hinting at a practical approach for the optimisation of the overall performance of heat exchanger articles. An X-Z specimen orientation shows the highest values of yield and ultimate strength and elongation at break at a Young's modulus of approximately 70 GPa. in situ SEM studies of tension response and corresponding DIC analysis are particularly suitable to highlight the peculiarities of mechanical behaviour, such as unevenness of strains and their localisation in the vicinity of the ultimate crack. The complexity of the processes (solidification at different cooling rates, frequency of reheating events, natural and artificial aging) forming the final structure requires fundamental characterisation research to guide the optimisation of overall performance in the most efficient way.

**Supplementary Materials:** The following are available online at https://www.mdpi.com/2227-7080/9/1/21/s1, Videos S1–S6: Fracture process videos of specimens with different printing orientations.

**Author Contributions:** Conceptualisation, D.R. and K.V.N.; methodology, K.V.N. and E.S.S.; software, E.S.S.; validation, D.R., A.I.S. and A.M.K.; formal analysis, E.S.S. and A.I.S.; investigation, E.S.S.; resources, A.I.S.; data curation, E.S.S.; writing—original draft preparation, D.R., E.S.S., A.I.S., and A.M.K.; writing—review and editing, E.S.S., A.I.S., and A.M.K.; visualisation, E.S.S.; supervision, A.I.S. and A.M.K.; project administration, D.R.; funding acquisition, D.R. All authors have read and agreed to the published version of the manuscript.

**Funding:** This research received no external funding.

**Institutional Review Board Statement:** Not applicable.

**Informed Consent Statement:** Informed consent was obtained from all subjects involved in the study.

**Data Availability Statement:** The data presented in this study are available in the Supplementary Materials.

**Acknowledgments:** The authors thank Julia Malakhova for the sample preparation.

**Conflicts of Interest:** The authors declare no conflict of interest.

**Appendix A**

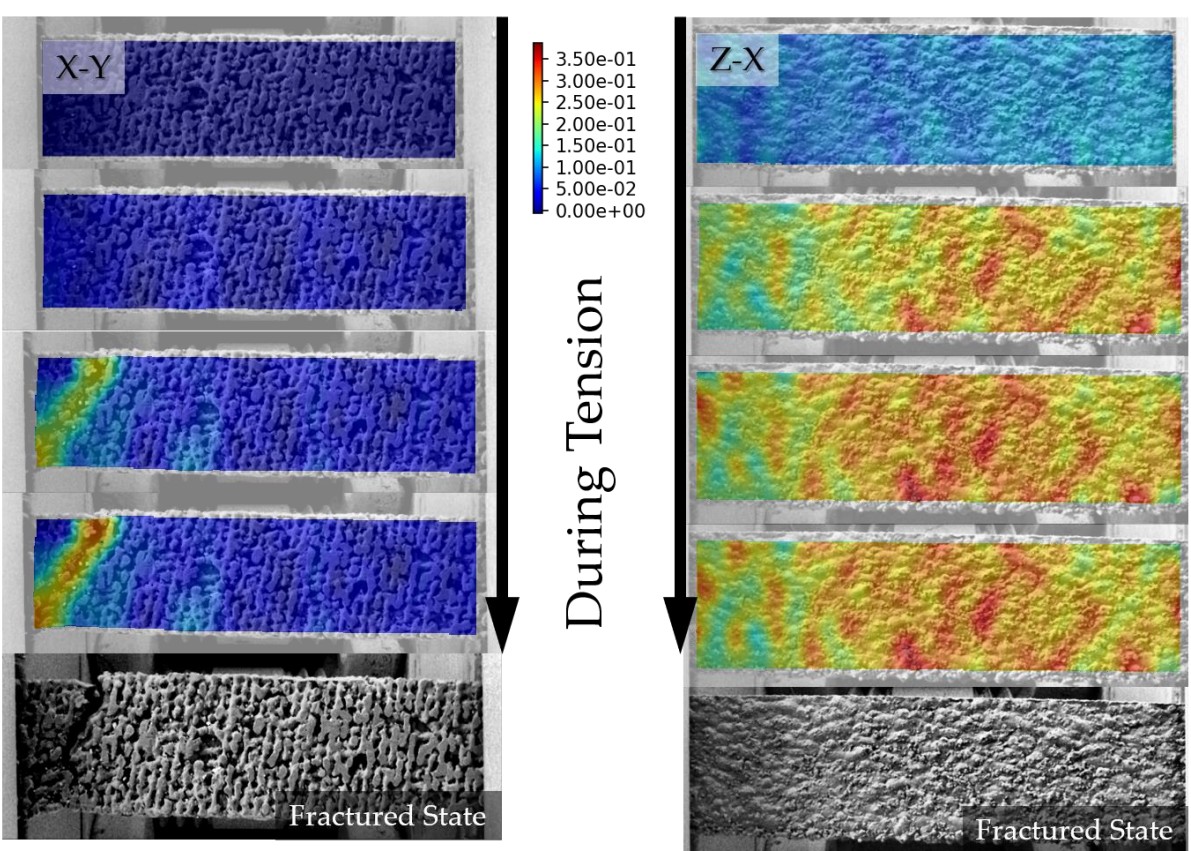

**Figure A1.** 2D strain distributions along loading axis for samples with different printing orientations: (**left**) X-Y and (**right**) Z-X.

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
