# Peer review of "In Situ SEM Study of the Micro-Mechanical Behaviour of 3D-Printed Aluminium Alloy"

_technologies, doi:10.3390/technologies9010021_

Round 1

Reviewer 1 Report

It seems that the discussion with results obtained by different authors for similar materials should be deeper. 

Author Response

Authors would like to thank the reviewers for the detailed comments and suggestions for the manuscript. We believe that the comments have identified important areas which required improvement. After completion of the suggested edits, the revised manuscript has benefitted from an improvement in the overall presentation and clarity. Below, you will find a point-by-point description of how each comment was addressed in the manuscript.

Reviewer 2 Report

Attached!

Author Response

(The authors gave the same response as above.)

Reviewer 3 Report

The article presents the results of research on the properties of powder incrementally formed elements using a laser beam. Strength samples were made by printing and subjected to tensile stress after heat treatment (aging) with simultaneous observation using SEM and DIC analysis.

Printing was performed in 3 directions with respect to the sample axis. The obtained results are interesting and I think they are extremely useful for predicting the properties of products made on the basis of incremental manufacturing technologies. The tests used DN 50 powder with a chemical composition corresponding to the AlSi3Mg0.5 alloy. The conducted in situ tests revealed a significant influence of the element orientation on the mechanical properties of the alloy. Additionally, they confirmed the lack of plasticity of the material (narrowing) during stretching, which should be attributed to the material properties.

Nevertheless, the material itself reveals the presence of unfused powder (line 217) which is visible on the surface of the element but not on the metallographic examination and the fracture. How can you explain it?

In addition, what was the temperature during the preparation of the sample, was it not high enough to cause the material to annealing? It seems that the thickness of the single layers is so small that they should not significantly affect the residual stress, more than less the strength of the sample (lines 282-287).

Author Response

(The authors gave the same response as above.)
